# Imputation strategies for missing baseline neurological assessment covariates after traumatic brain injury: A CENTER-TBI study

Ari Ercole[1,2]*, Abhishek Dixit[1], David W. Nelson[3], Shubhayu Bhattacharyay[1], Frederick A. Zeiler[4], Daan Nieboer[5], Omar Bouamra[6], David K. Menon[1], Andrew I. R. Maas[7], Simone A. Dijkland[5,8], Hester F. Lingsma[5,8], Lindsay Wilson[9], Fiona Lecky[10], Ewout W. Steyerberg[5,8,11], the CENTER-TBI Investigators and Participants[¶]

**1** Division of Anaesthesia, University of Cambridge, Cambridge, United Kingdom, **2** Centre for Artificial Intelligence in Medicine, University of Cambridge, Cambridge, United Kingdom, **3** Section for Anesthesiology and Intensive Care, Department of Physiology and Pharmacology, Karolinska Institutet, Stockholm, Sweden, **4** Section of Neurosurgery, Department of Surgery, Rady Faculty of Health Sciences, University of Manitoba, Winnipeg, Canada, **5** Department of Public Health, Erasmus University Medical Center, Rotterdam, Netherlands, **6** Trauma Audit Research Network, University of Manchester, Salford, United Kingdom, **7** Department of Neurosurgery, Antwerp University Hospital and University of Antwerp, Edegem, Belgium, **8** Center for Medical Decision Making, Erasmus University Medical Center, Rotterdam, Netherlands, **9** Division of Psychology, University of Stirling, Stirling, United Kingdom, **10** Centre for Urgent and Emergency Care Research (CURE), School of Health and Related Research (ScHARR), University of Sheffield, Sheffield, United Kingdom, **11** Department of Medical Statistics and Bioinformatics, Leiden University Medical Center, Leiden, Netherlands

¶ Membership of the CENTER-TBI investigators and participants is listed in the Acknowledgments.
* ae105@cam.ac.uk

**Data Availability Statement:** The data analysed in this study is third party data that can be obtained on application to the CENTRE-TBI management

## Abstract

Statistical models for outcome prediction are central to traumatic brain injury research and critical to baseline risk adjustment. Glasgow coma score (GCS) and pupil reactivity are crucial covariates in all such models but may be measured at multiple time points between the time of injury and hospital and are subject to a variable degree of unreliability and/or missingness. Imputation of missing data may be undertaken using full multiple imputation or by simple substitution of measurements from other time points. However, it is unknown which strategy is best or which time points are more predictive. We evaluated the pseudo-$R^2$ of logistic regression models (dichotomous survival) and proportional odds models (Glasgow Outcome Score—extended) using different imputation strategies on the The Collaborative European NeuroTrauma Effectiveness Research in Traumatic Brain Injury (CENTER-TBI) study dataset. Substitution strategies were easy to implement, achieved low levels of missingness (<< 10%) and could outperform multiple imputation without the need for computationally costly calculations and pooling multiple final models. While model performance was sensitive to imputation strategy, this effect was small in absolute terms and clinical relevance. A strategy of using the emergency department discharge assessments and working back in time when these were missing generally performed well. Full multiple imputation had the advantage of preserving time-dependence in the models: the pre-hospital assessments were found to be relatively unreliable predictors of survival or outcome. The predictive performance of later assessments was model-dependent. In conclusion, simple substitution

committee at https://www.center-tbi.eu/. Information on how to request data, including data access request forms, can be found here: https://www.center-tbi.eu/data. The authors had no special access privileges for this study.

**Funding:** CENTER-TBI is supported by The European Union FP 7th Framework program (grant 602150) with additional funding provided by the Hannelore Kohl Foundation (Germany), Integra Life Sciences and by the non-profit organization One Mind For Research (directly to INCF). SB is currently funded by a Gates Cambridge fellowship. FAZ is supported by the University of Manitoba VPRI Research Investment Fund (RIF), Winnipeg Health Sciences Centre (HSC) Foundation, and the University of Manitoba Rudy Falk Clinician-Scientist Professorship. The funders had no role in study design, data collection and analysis, decision to publish, or preparation of the manuscript.

**Competing interests:** The authors have declared that no competing interests exist.

strategies for imputing baseline GCS and pupil response can perform well and may be a simple alternative to full multiple imputation in many cases.

## Introduction

As a major worldwide cause of death, disability and socioeconomic burden [1], traumatic brain injury (TBI) continues to be an important area of research into novel or better stratified interventions or systems of care. Baseline risk adjustment is critical to any outcome-focused research project aiming to better understand the influence of putative factors or treatments that may improve outcomes. There have been a number of attempts at baseline characterisation of which the International Mission on Prognosis and Clinical Trial Design in Traumatic Brain Injury (IMPACT) [2] and Corticosteroid Randomisation After Significant Head Injury (CRASH) [3] model are perhaps the best known. In common with other such models the presenting Glasgow coma score (GCS- or its motor component GCSm) and number of unreactive pupils are both important clinical parameters describing injury severity at presentation.

There is some potential for ambiguity in defining the baseline GCS and pupil reactivity as both may be confounded by other contributors to unconsciousness such as hypoxia or hypotension before resuscitation. The pre-hospital and resuscitation phase of severely injured TBI patients is a high-pressure and time-critical period which may additionally involve handover of care between a number of different individuals, not all of whom may be healthcare professionals experienced in GCS assessment. Coupled with a general lack of documentation standardisation between pre-hospital and hospital care, there may be a high proportion of such early data that will be missing, unreliable or confounded by under-resuscitation [4]. Dealing with these missing covariates is a critical data curation task since baseline adjustment underpins almost all analyses and therefore deserves particular methodological attention.

Complete case analysis is generally statistically undesirable and therefore some form of imputation strategy will be necessary to deal with missing values. Arrival [5] or post-emergency resuscitation scores may be substitutes, but it is not clear which is best or, when more than one is present, which to use. Furthermore, the GCS after TBI is not static; instead it may evolve dramatically in the early phase, and, since it is impossible to define standardised timepoints given the vast heterogeneity of TBI presentations, this may influence the measured level of consciousness even between similarly injured patients. There is no clear consensus on which time points to use; for example, IMPACT and Trauma Audit and Research Network (TARN) models [6] have published different approaches.

The concept of a 'post-stabilisation' score (as used in IMPACT) is superficially appealing but also problematic to unabiguously define clinically. In modern pre-hospital care and emergency medicine systems, anaesthesia and endotracheal intubation are key interventions which may even occur simultaneously with other resuscitation efforts so that the motor and verbal GCS sub-scores may be unavailable in the 'best resuscitated'/'post stabilisation' case. Again, this mandates some form of imputation. One simple 'shorthand' approach used by clinicians is to rate these components a '1' but this approach may over-estimate injury severity. Some form of imputation is desirable, and, whilst a regression model for the estimation of missing values has been published [7], such a univariate approach does not include other covariates and also discards discriminating pre-intubation information which might be prognostically important.

Multiple imputation is perhaps the most appropriate approach for dealing with missingness when constructing prediction models [8] and is likely to be less biased than complete case analysis if performed carefully. However, the computation of many large datasets may be computationally expensive and reproducibility will be determined by choice of covariates for the imputation as well as the imputation method chosen and the random seed. Multiple imputation offers the advantage of creating a complete dataset of covariates, avoiding combining data across various time-points. However, the variance from the various imputations must be incorporated in some way into the uncertainty of the final estimator. Methods for pooling results across imputations exist for many simpler model types but may not be established in other cases in general (in which case researchers may need to instead use a single imputation or alternative strategy).

The Collaborative European NeuroTrauma Effectiveness Research in Traumatic Brain Injury (CENTER-TBI) study [9, 10] is a large, pan-European observational study which aims to better understand the determinants of outcome and optimal treatment by better clinical phenotyping through deep data collection. Detailed data on early neurological assessments as well as outcome data (including, *inter alia*, the extended Glasgow outcome score—GOSE) was collected. For all the reasons above there is considerable uncertainty in how best to robustly define the most 'reliable' GCS (or sub-score) to use in such observational research and what difference different imputation assumptions might make. It is therefore likely to be critical to establish this to ensure a consistent, principled and reproducible approach, and such considerations will be equally appropriate to other studies.

The CENTER-TBI dataset is particularly complex. The GCS and pupil response variables in CENTER-TBI are recorded at several time points: pre-hospital, arrival at any referring hospital (for secondary transfers), arrival at study hospital and post-stabilization. The objective of this work was to determine a clinically and statistically plausible method to obtain a derived baseline GCS for prognostic analyses in CENTER-TBI. We investigated different substitution methods for dealing with missing GCS and pupil reactivity data as well as comparing these strategies to more numerically cumbersome multiple imputation. Furthermore, we set out to determine the effect of different strategies for imputing missing motor and verbal sub-scores in anaesthetised and ventilated patients.

## Materials and methods

We assessed the performance of these strategies and their combinations by comparing McFadden's pseudo-$R^2$ for both logistic regression (for dichotomous alive/dead outcome) and proportional odds logistic regression (for the ordered categorical modelling of GOSE) using the other IMPACT predictors as additional covariates.

### Patients and data

Like many TBI studies, the IMPACT model was based on patients with moderate and severe TBI (i.e., GCS $\leq$ 12). By contrast, the CENTER-TBI study is instead stratified into 'emergency department', 'hospital admission', 'ICU admission' strata. These strata are not directly comparable to the traditional 'mild', 'moderate' and 'severe' categorization. Since re-stratification would require the use of an optimal GCS, which is itself the goal of this work, we instead created models for all patients and for the ICU stratum only to avoid this circular logic. The latter stratum is likely to best approximate combined moderate/severe categories modelled in IMPACT.

We used release 1.0 of the CENTER-TBI data set with local data hosting, management and extraction on the Opal platform [11]. GCS component and number of unreactive pupils data

from the original data set were available for the following time-points: pre-hospital, arrival at referral centre ED (where a secondary transfer took place), arrival at study hospital and ED discharge ('post-stabilisation'). IMPACT covariates were extracted from the electronic case report form data.

## Ethics

The study was authorised by the CENTER-TBI management committee. The ethical approvals for the CENTER-TBI study have been previously described [9, 10] and a full list of ethical approvals is available at https://www.center-tbi.eu/.

## Imputation by substitution strategies

We considered five approaches for obtaining a derived GCS, GCS motor score and pupil reactivity:

- 'IMPACT' approach: start with ED discharge assessments (approximates the 'post-stabilisation' score used in IMPACT). If absent, substitute with the next available value going back in time. I.e., ED discharge → study hospital ED arrival → referring hospital ED arrival → pre-hospital.

- 'TARN' approach: start with the value on arrival to the referring hospital ED. If absent, find most reliable score going forward in time. I.e., referring hospital ED arrival (for secondary transfers) → study hospital ED arrival. If this is missing then the prehospital score is used.

- 'Best score' approach (to reflect the severity of primary injury without neuroworsening/once resuscitated): Best neurological status (highest GCS sum or GCS motor score, fewest unreactive pupils) across all time points (pre-hospital/referring hospital (secondary transfers)/study hospital ED arrival/post-stabilization).

- 'Erasmus' approach: start with the study hospital arrival GCS score (reflects complete pre-hospital/referring centre stabilisation). If this is missing, use sequence study hospital arrival → referring hospital arrival (for secondary transfers) → pre-hospital.

- 'Worst score' approach (most pessimistic assessment): Worst neurological status (lowest GCS sum or GCS motor score, most unreactive pupils across all time points.

The outcome of interest was 6 month survival or GOSE. Where this was missing, it was imputed (as explained below).

## Statistical analysis

We used the same predictors as were found to be significant in the IMPACT model, viz. we added candidate GCS/GCSm and pupil unreactivity to a set of 'fixed' baseline characteristics; age, glucose, haemoglobin and CT characteristics (Marshall score, the presence of traumatic subarachnoid haemorrhage or epidural haematoma). We used 6 month GOSE as our outcome of interest.

To evaluate 'simple' imputation by substitution strategies for GCS, GCSm and pupil reactivity it was necessary to first obtain a fully imputed set of the remaining covariates and outcomes so that we did not have additional missingness in these parameters that varied between models. Linear regression was used for imputation of haemoglobin and glucose which were approximately normally distributed. Logistic regression was used for dichotomous presence or absence of an extradural haematoma or traumatic subarachnoid haemorrhage. Marshall score and 6 month GOSE (the outcome of interest) was imputed using a proportional odds model

which also included 3 and 12 month GOSE for the imputation to better approximate 6 month GOSE. Cases in which no GOSE was available were deleted after imputation for statistical efficiency. Age, hypoxia and hypotension were included in the imputation as predictors but did not need to be imputed themselves: age data was complete and hypoxia and hypotension were assumed to not have been present if the data were missing.

The putative GCS, GCSm and pupil reactivity scores were then calculated as described above using Opal. The similarity among these scores was compared by pairwise Spearman's correlation analysis. We then used proportional odds logistic regression to build models with 6 month GOSE as an ordinal outcome evaluating all combinations of GCS or GCSm and pupil versions. We also constructed equivalent logistic regression models for 6 month survival status. Model performance was compared using their (McFadden's) pseudo-$R^2$ values.

To evaluate the performance of models with a completely imputed data set, similar imputations were performed for the complete data set by also including the GCS and GCS motor score as well as pupils for pre-hospital, study or referring hospital ED arrival score and ultimate ED discharge time points. Because the 'referring' ED time point applies only to the subset of patients who underwent a secondary transfer to the study centre, the imputed 'referring' and 'study hospital' arrival assessments were then combined into a single 'presenting ED arrival' time point representing the neurological assessment at the first ED to which the patient presented.

In all cases, 200 imputed data sets were obtained using 5 iterations using the MICE (version 3.3.0) package [12] for the R statistical programming language [13]. MASS (version 7.3.50) was used to build the proportional-odds model. The imputations were run on the machine (Intel Xeon Silver 4110 2.1G, 8C/16T, 10 GB RAM, 240 GB Hard disk) running Ubuntu 16.04.5 LTS (Xenial). Cases with no GOSE at any of the time points were included in the imputation but subsequently deleted before modelling for statistical efficiency in accordance with the methodology of [14]).

## Results

The patient characteristics of the CENTER-TBI data set will be published elsewhere. However, for all strata there were ($n = 4, 509$) patients and the ICU stratum subset consisted of ($n = 2, 138$) patients. Multiple imputations each took approximately 6 hours of dedicated machine time.

### Imputation by substitution

The above substitution strategies were implemented and the percentage missing for different imputation by substitution strategies for GCS and GCSm are shown in Table 1.

Fig 1 shows how the resulting imputed data sets using different imputation strategies are weighted across time points. The distribution of GCS between 'mild', 'moderate' and 'severe' categories was similar and is presented in the Supporting information for all strata and the ICU stratum (S1 and S2 Figs). The GCS and GCSm and also pupils were highly correlated between imputation strategies (Fig 2) illustrating their degree of equivalence.

Fig 3 shows the variation of McFadden's pseudo-$R^2$ for logistic regression models for alive/dead constructed using differing 'simple' imputation by substitution schemes and using different strategies for handling motor and verbal scores for intubated or sedated patients. There was a variation in pseudo-$R^2$ between models that was often statistically significant but small in absolute value. For all stratum models, the optimal model (with $R^2 \approx 0.44$) was obtained by using the full GCS sum score and using the IMPACT methodology for imputation (although combining IMPACT GCS with ERASMUS pupils performed marginally better still) with

**Table 1. Percent missing data for GCS sum score, GCS motor component and pupil reactivity for different imputation by substitution strategies (all strata).**

|  | BEST | WORST | IMPACT | TARN | ERASMUS |
|---|---|---|---|---|---|
| **GCS sum score** | | | | | |
| NC if missing either v or m | 3.97 | 3.97 | 3.97 | 5.74 | 5.74 |
| NC if missing m, missing v = 1 | 2.77 | 2.77 | 2.77 | 4.72 | 4.72 |
| NC if missing v, missing m = 1 | 3.95 | 3.95 | 3.95 | 5.70 | 5.70 |
| Missing v or m = 1 | 2.46 | 2.46 | 2.46 | 4.37 | 4.37 |
| **GCS motor score** | | | | | |
| NC if missing m | 2.48 | 2.48 | 2.48 | 4.46 | 4.46 |
| Missing m = 1 | 2.02 | 2.02 | 2.02 | 3.90 | 3.90 |
| **Pupil reactivity** | | | | | |
| Missing pupil response | 5.81 | 5.81 | 5.81 | 8.56 | 8.56 |

For the sum score, where either or both of the GCS verbal of motor scores were missing (i.e., if endotracheally intubated or when under deep sedation/paralyzed respectively), the GCS was either not calculated (NC) or calculated by substituting 1 for either m or v according to the logic above. For the motor component, the effect of either not calculating (NC) a score or replacing this with 1 is presented. Data is for all strata.

deletion of patients where motor or verbal subscores could not be assessed. For the ICU stratum only (S3 Fig), the optimal strategy ($R^2 \approx 0.38$) was to use the best-neurology motor score although the best-neurology pupils was consistently sub-optimal across the models. In this group, treating un-assessed/untestable verbal or motor subscores as '1' produced comparable or possibly marginally better models than a deletion strategy.

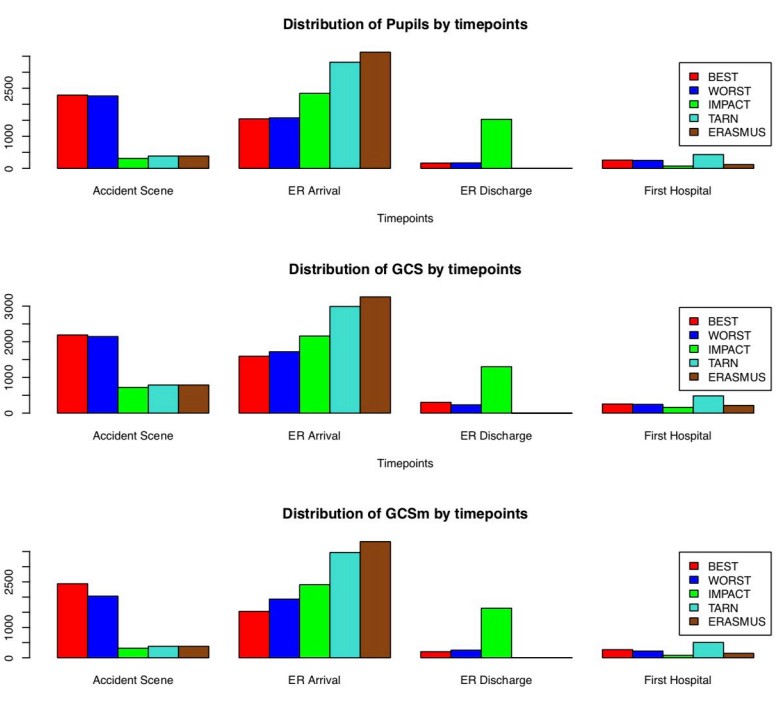

**Fig 1. Distribution of data choices across time points by imputation by substitution strategy reflecting completeness of the underlying data.** For example, although the ERASMUS and IMPACT approaches are heavily dominated by data from the ED arrival, the IMPACT method is also weighted by a similar proportion of assessments made at ED discharge. In contrast, the best and worst neurological assessments were weighted towards earlier, pre-hospital and ED arrival, time points.

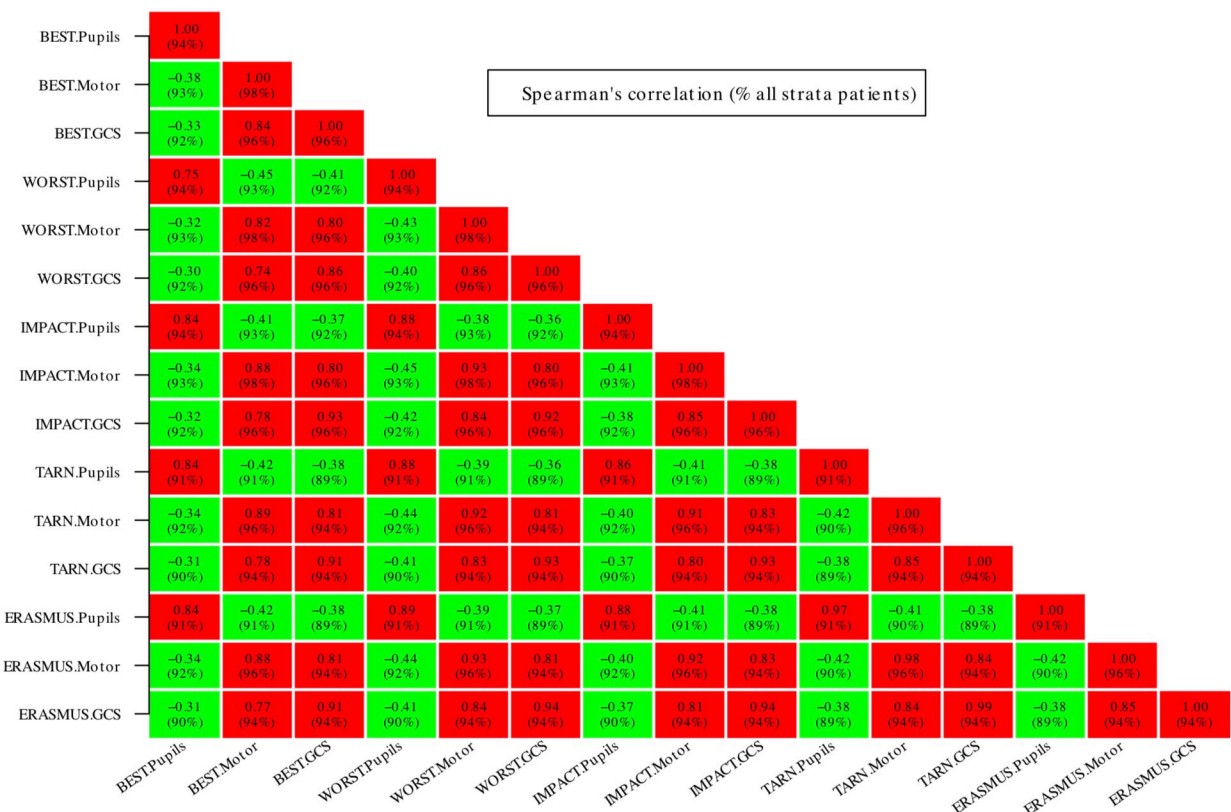

**Fig 2. Spearman's correlations between imputed data for all patients (complete case analysis with verbal/motor scores for intubated/sedated patients treated as missing).** The figure shows that GCS, GCSm and pupil reactivity variables are highly correlated between substitution imputation methods. Furthermore, the GCS sum and motor component are also highly correlated. There is a modest negative correlation between imputed versions of pupil reactivity and GCS sum or motor score (the negative correlation reflecting the coding of pupils as the number of *unreactive* pupils).

The equivalent results for proportional odds models for GOSE are shown in Fig 4. Again, there was variability in model performance that was typically statistically significant but small. Values of $R^2 \approx 0.16$ were much lower than for the dichotomous/logistic models. For the all strata models, the optimal model was obtained by combining the IMPACT method full GCS sum scores with either IMPACT or ERASMUS imputed pupil responses. In this group, imputing unavailable motor component as 1 but deleting cases for which verbal component was not assessable gave the best model (with $R^2 \approx 0.165$). For the ICU stratum, the best model (with $R^2 \approx 0.14$) was obtained using the best-neurology full GCS sum and by imputing missing motor and verbal components as 1 (S4 Fig). There was little to choose between pupil imputation strategies except that the best neurology strategy performed consistently badly.

## Full multiple imputation models

Fig 5 compares the performance of logistic regression (alive/dead) models using GCS and pupil time points that are fully imputed. For the all strata group, the optimal strategy for handling missing motor or verbal components was deletion and imputation. Model performance was typically better using the GCS sum rather than motor sub-score and there was a consistent trend to better model performance using neurological assessments from later time points (although using pupil assessments at ED discharge was marginally inferior). The best possible $R^2$ was $\approx 0.44$. For the ICU group, deleting and imputing missing motor or verbal GCS time

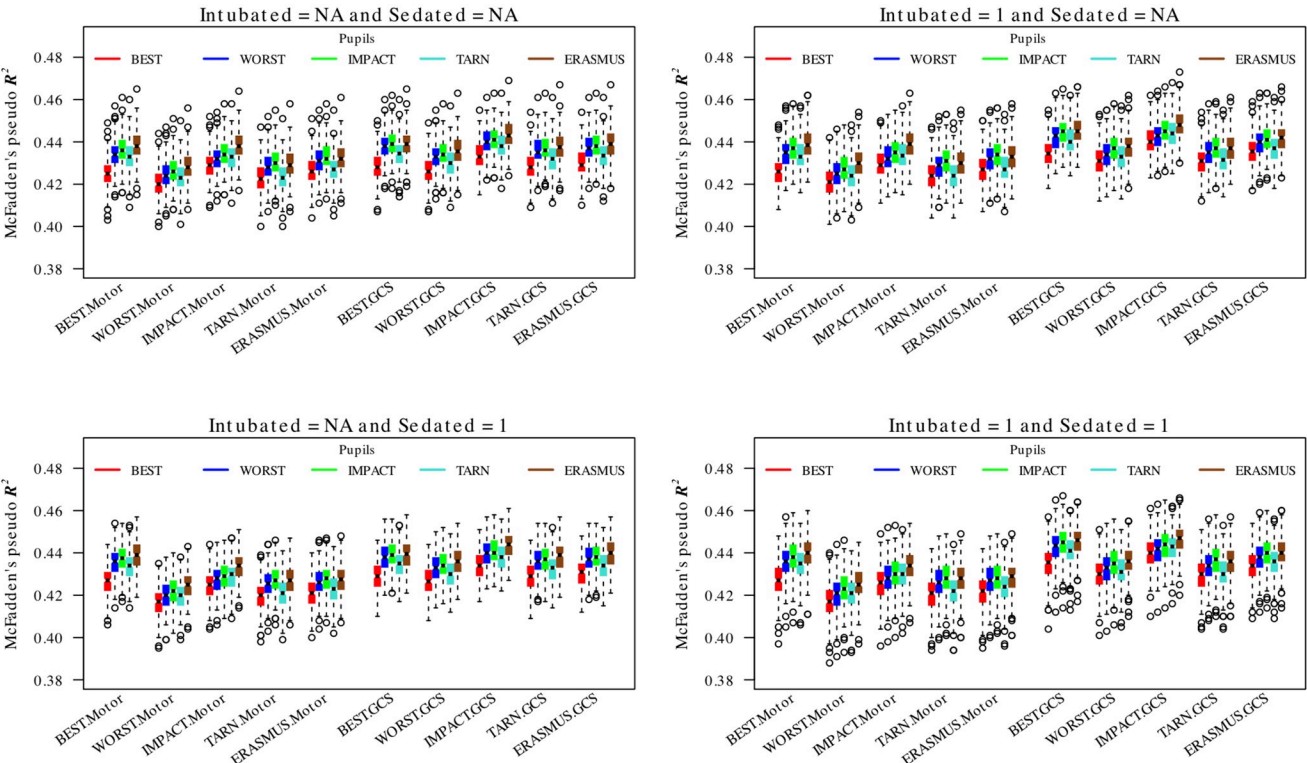

**Fig 3. Comparison of logistic regressions for dichotomous 6 month survival/death for different combinations of GCS, GCS-motor score and pupil response imputation choices.** The boxes/whiskers reflect the variability from the 200 imputed data sets used. Data shown for all strata (data for the ICU stratum was similar and is presented in the S3 Fig).

points was also a good strategy, yet substituting missing motor as 1 was marginally better still. Use of the full GCS sum again marginally outperformed the models based on the motor component alone. A similar consistent pattern to improved models at later time points (again, with ED discharge pupils performing marginally worse than the ED arrival values). For the ICU group (S5 Fig), the best $R^2 \approx 0.38$. Models based on pre-hospital neurological assessments were the worst performing for both all strata and ICU stratum groups.

The results for the fully-imputed proportional odds models are shown in Fig 6. Again, model performance was lower than that of the logistic regression model overall but slightly higher for the all strata group (best $R^2 \approx 0.155$) compared to the ICU subgroup (best $R^2 \approx 0.135$; S6 Fig). Deletion and imputation of missing motor and verbal sub-components was the best strategy in both groups. However, in contrast with the results of the logistic regression model, as summarised in Fig 5, the optimal time for prediction (in both groups) of the proportional odds model was at the time of arrival to the presenting hospital rather than at ED discharge. Again, using the pre-hospital time-point consistently generated the worst performing models.

## Discussion

These results from CENTER-TBI are broadly in agreement with the performance of those previously quoted [15]. It is important to point out that the (pseudo)variance is highly dependent on model choice (e.g., dichotomous alive/dead vs proportional odds model for ordered GOSE categories, which is a more difficult modelling task, statistically speaking and has a

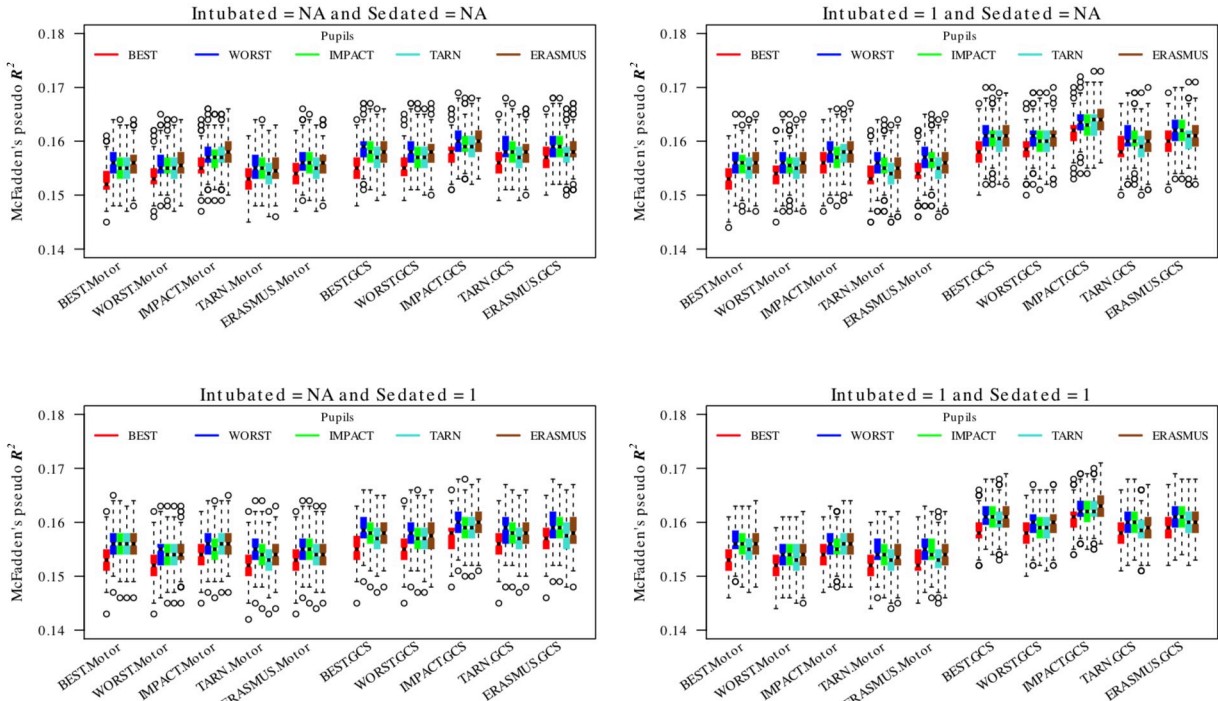

**Fig 4. Comparison of pooled proportional odd regressions for GOSE for different combinations of GCS, GCS-motor score and pupil response imputation choices.** The explanatory value of the prehospital time point is consistently limited. The boxes/whiskers reflect the variability from the 200 imputed data sets used. Data shown for all strata (data for the ICU stratum was similar and is presented in the S4 Fig).

commensurately smaller $R^2$). Similarly, it is perhaps not surprising that model performance varies between patient subgroups; models for all strata generally performed better than those for the ICU subgroup (see Supporting information for figures) although the difference in $R^2$ was much smaller than the difference between the choice of logistic or proportional odds models. This is presumably because the all strata group contains a proportionately higher contribution from less severely injured patients who are more homogeneous.

Our results also demonstrate a sensitivity of model performance to choice of imputation strategy. This sensitivity is typically statistically significant (even accounting for the very large number of comparisons in this work) but small in absolute $R^2$ terms. Therefore for applications, such as baseline risk adjustment, where a truly optimal $R^2$ is not of primary importance, there is little to choose between these strategies. Broadly speaking, using the full GCS sum performs better than the motor score alone. Furthermore, a very simple imputation by substitution strategy based on filling in missing values using other available time-points outperforms full multiple imputation and is computationally far more straightforward. The IMPACT strategy (working backwards from the ED discharge) seems to generally work well, though 'best GCS' and 'best GCSm' approaches perform extremely well on the ICU stratum subset of patients presumably because these parameters best reflect the degree of primary neurological injury. In contrast, however, use of the best (i.e., highest number of reactive) pupils was a consistently poor strategy. This is interesting and presumably reflects a comparatively disproportionate effect of even an episode of one or two unreactive pupils on outcome.

For applications (e.g., in generating a prediction tool) where optimal model performance is important, the sensitivity to imputation strategy may be more important, and it is likely that the optimal choice will depend on the details of the task at hand. Furthermore, the inclusion

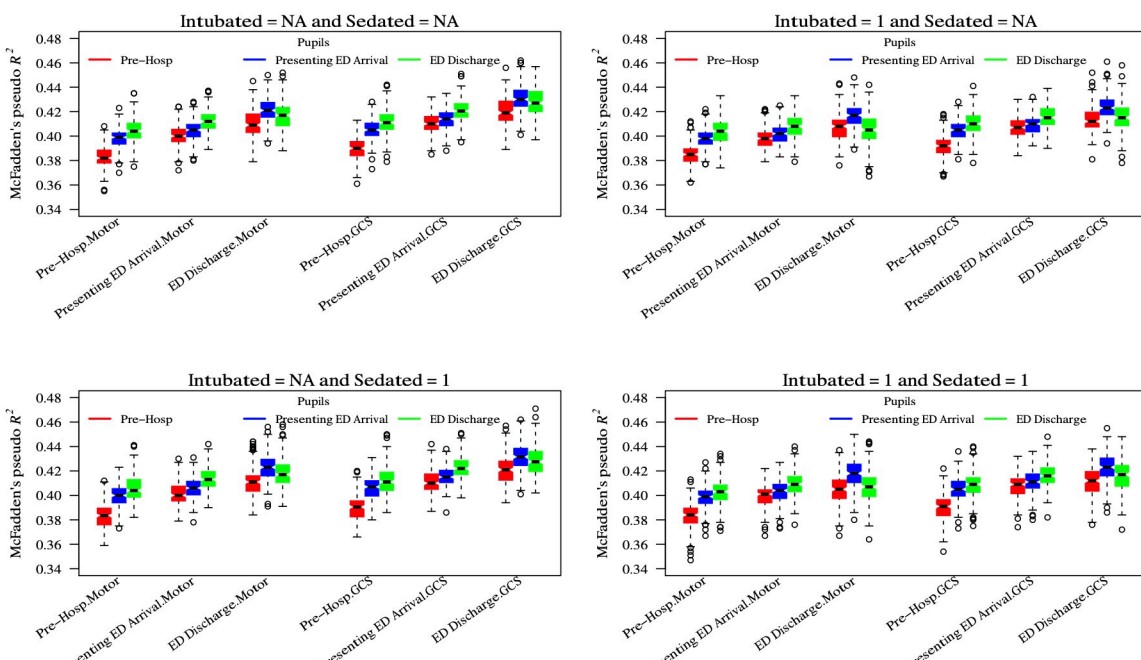

**Fig 5. Comparison of pooled logistic regressions for 6 month GOSE for fully imputed time points.** The 'presenting' ED arrival time point was a composite formed from the 'referring' and 'study hospital' ED time points to reflect the first contact with the ED irrespective of whether the patient underwent secondary transfer. The boxes/whiskers reflect the variability from the 200 imputed data sets used. Data shown for all strata (data for the ICU stratum was similar and is presented in the S5 Fig).

criteria, patient group and precise structure or missingness in the CENTER-TBI data set is likely to be different from other data sets, and it is likely that our findings will not generalise. Therefore, for specific applications where high model performance is *mission critical*, we recommend that a bespoke evaluation, analogous to that presented here, is undertaken to ensure the best strategy is obtained.

We have demonstrated that simple strategies for imputation of of GCS/pupil reactivity can outperform formal multiple imputation in terms of $R^2$, simplicity of use and computing requirements. As demonstrated in Table 1, very low levels (<10%) can be achieved with such simple approaches, and therefore, imputation by substitution using, say, an 'IMPACT' approach has merit over a multiply-imputed strategy for general purposes. Having said this, such simple models achieve imputation by sacrificing the temporal data, and fully imputed models allow predictions at multiple time-points which may be of interest in some applications. Indeed, our findings from the multiply-imputed case are interesting. The models based on the pre-hospital time-point consistently perform worse than for those using later time points. This finding may be because pre-hospital clinical assessment is intrinsically less predictive with outcome being dominated by subsequent neuroworsening events. However, another equally plausible explanation is that this is simply a reflection of the high degree of missingness and variable accuracy of pre-hospital data. In either case, our results demonstrate the unreliability of pre-hospital data which is an important result when analysing studies or planning future studies.

It is interesting that there is a monotonic increase in logistic regression performance with neurological assessments at later time points being more predictive of outcome. Thus, the ED

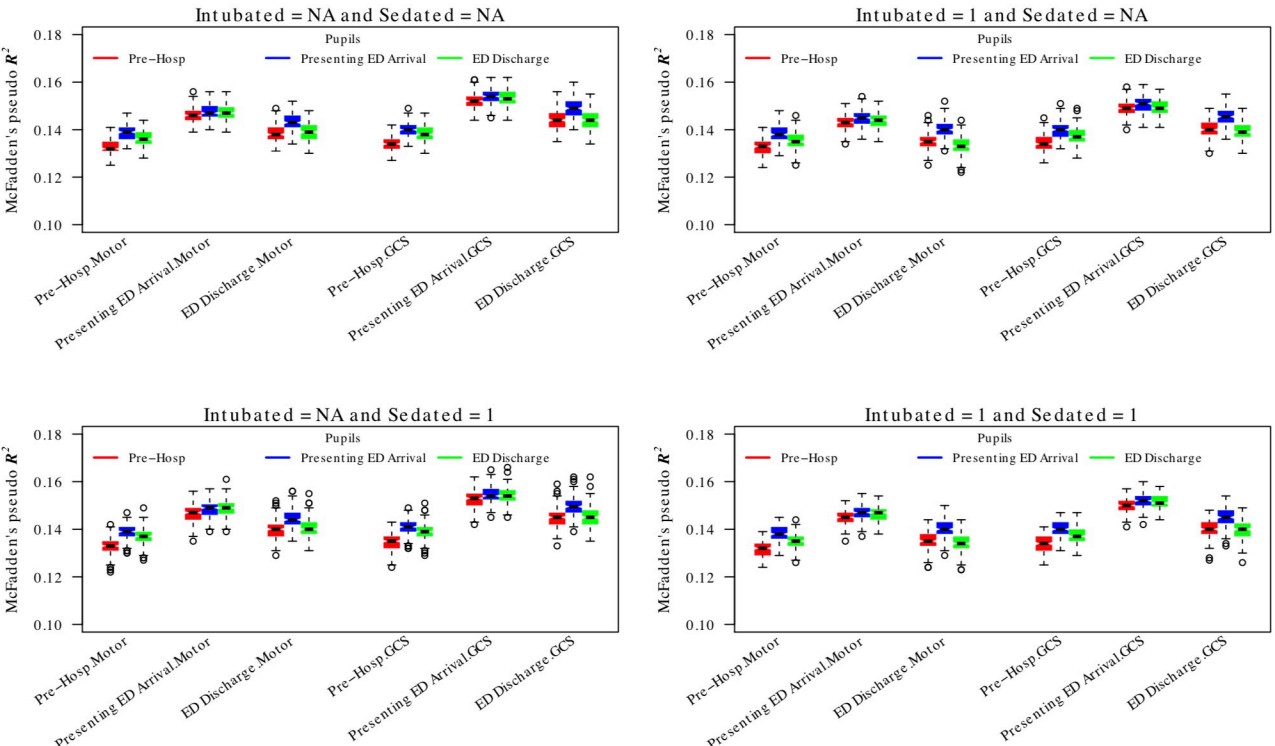

**Fig 6. Comparison of pooled proportional odds regressions for 6 month GOSE for fully imputed time points.** The 'presenting' ED arrival time point was a composite formed from the 'referring' and 'study hospital' ED time points to reflect the first contact with the ED irrespective of whether the patient underwent secondary transfer. The boxes/whiskers reflect the variability from the 200 imputed data sets used. Data shown for all strata (data for the ICU stratum was similar and is presented in the S6 Fig).

discharge time point is, broadly speaking, recommended from our findings (although the trend was perhaps a little less pronounced in the ICU subgroup). This is perhaps not surprising since ED discharge is temporally closest to outcome and occurs after a period where the patients are maximally 'differentiated'. However, we have demonstrated that this is also model-specific. When a proportional odds model is used to model GOSE instead, the ED discharge time point performs almost as poorly as the pre-hospital one, and predictions are better based on the neurology at arrival at the presenting hospital. It is hard to be certain why this might be but clearly there is some factor that is predictive of intermediate outcomes but not gross survival that is diluted by the time of ED discharge. It is noteworthy that that ED discharge methods are heavily weighted towards ED arrival GCS measures, as illustrated in Fig 1. Again, this reinforces the need to individualise the time points of neurological assessment when planning TBI studies.

It is important to stress that what we have herein referred to as the 'TARN' imputation refers only to the order in which we choose substitutes for this 'simple' imputation. The actual TARN model uses a more sophisticated range of covariates (with GCS polytomised to reflect predictive power and additional categories to reflect injury severity and intubation status). In this work we have instead used the IMPACT model structure for all evaluations. This facilitates direct comparison as we are not interested necessarily in finding a new model but instead in comparing the predictive information content of different GCS/pupil response formulations. However, this may account for the slightly poorer performance of the TARN imputation in absolute terms; optimum imputation strategies for other bespoke models, such as that of

TARN, would need to be evaluated on a case-by-case basis if achieving the highest possible $R^2$ were of importance.

Our analysis was conducted on data from the CENTER-TBI study in order to determine the optimal imputation strategy in this case. Other studies are likely to have differing patient demographics, and the numbers and structural determinants of patterns of missingness may well differ in alternative medical systems. Indeed the patient populations we used are not directly comparable to the IMPACT group as the latter was developed on a group of moderate and severe TBI patients whereas CENTER-TBI was designed to be stratified by ICU, hospital or ED admission instead. As a result, we cannot be sure that our results will generalise directly. However, we believe that we have presented a statistical framework which other studies may wish to follow as a first step when considering their best baseline risk adjustment variables.

## Conclusions

Simple imputation of missing GCS, GCS motor score or pupil reactivity using substitution strategies are computationally trivial and can outperform full multiple imputation in terms of higher pseudo-$R^2$ and achieve data missingness of less than 10%. For both logistic regression survival and proportional odds GOSE models, the $R^2$ is sensitive to imputation strategy and patient subgroup. However, this variability is small in absolute terms, and a strategy based on using assessments at discharge from the ED and choosing earlier time points if these are missing generally performs well. Full imputation using chained equations may be useful if temporally-sensitive predictions are to be made. In this case, pre-hospital assessments are unreliable predictors.

## Supporting information

**S1 Fig. GCS distribution between 'mild', 'moderate' and 'severe' categories for all strata.**
(TIF)

**S2 Fig. GCS distribution between 'mild', 'moderate' and 'severe' categories for the ICU stratum.**
(TIF)

**S3 Fig. Comparison of logistic regressions for dichotomous 6 month survival/death for different combinations of GCS, GCS-motor score and pupil response imputation choices.** The boxes/whiskers reflect the variability from the 200 imputed data sets used. Data shown for the ICU stratum.
(TIF)

**S4 Fig. Comparison of pooled proportional odd regressions for GOSE for different combinations of GCS, GCS-motor score and pupil response imputation choices.** The explanatory value of the prehospital time point is consistently limited. The boxes/whiskers reflect the variability from the 200 imputed data sets used. Data shown for the ICU stratum.
(TIF)

**S5 Fig. Comparison of logistic regressions for dichotomous 6 month survival/death for different combinations of GCS, GCS-motor score and pupil response imputation choices.** The boxes/whiskers reflect the variability from the 200 imputed data sets used. Data shown from the ICU stratum.
(TIF)

**S6 Fig. Comparison of pooled proportional odds regressions for 6-month GOSE for fully imputed time points.** The 'presenting' ED arrival time point was a composite formed from

the 'referring' and 'study hospital' ED time points to reflect the first contact with the ED irrespective of whether the patient underwent secondary transfer. The boxes/whiskers reflect the variability from the 200 imputed data sets used. Data shown for the ICU stratum.
(TIF)

## Acknowledgments

### CENTER-TBI investigators and participants

Adams Hadie[1], Alessandro Masala[2], Allanson Judith[3], Amrein Krisztina[4], Andaluz Norberto[5], Andelic Nada[6], Andrea Nanni[2], Andreassen Lasse[7], Anke Audny[8], Antoni Anna[9], Ardon Hilko[10], Audibert Gérard[11], Auslands Kaspars[12], Azouvi Philippe[13], Baciu Camelia[14], Bacon Andrew[15], Badenes Rafael[16], Baglin Trevor[17], Bartels Ronald[18], Barzó Pál[19], Bauerfeind Ursula[20], Beer Ronny[21], Belda Francisco Javier[16], Bellander Bo-Michael[22], Belli Antonio[23], Bellier Rémy[24], Benali Habib[25], Benard Thierry[24], Berardino Maurizio[26], Beretta Luigi[27], Beynon Christopher[28], Bilotta Federico[16], Binder Harald[9], Beqiri Erta[14], Blaabjerg Morten[29], Borgen Lund Stine[30], Bouzat Pierre[31], Bragge Peter[32], Brazinova Alexandra[33], Brehar Felix[34], Brorsson Camilla[35], Buki Andras[36], Bullinger Monika[37], Bučková Veronika[33], Calappi Emiliana[38], Cameron Peter[39], Carbayo Lozano Guillermo[40], Carise Elsa[24], Carpenter K[41], Castaño-León Ana M[42], Causin Francesco[43], Chevallard Giorgio[14], Chieregato Arturo[14], Citerio Giuseppe[44,45], Cnossen Maryse[46], Coburn Mark[47], Coles Jonathan[48], Cooper Jamie D.[49], Correia Marta[50], Covic Amra[51], Curry Nicola[52], Czeiter Endre[53], Czosnyka Marek[54], Dahyot-Fizelier Claire[24], Damas François[55], Damas Pierre[56], Dawes Helen[57], De Keyser Véronique[58], Della Corte Francesco[59], Depreitere Bart[60], Ding Shenghao[61], Dippel Diederik[62], Dizdarevic Kemal[63], Dulière Guy-Loup[55], Dzeko Adelaida[64], Eapen George[15], Engemann Heiko[51], Ercole Ari[65], Esser Patrick[57], Ezer Erzsébet[66], Fabricius Martin[67], Feigin Valery L.[68], Feng Junfeng[61], Foks Kelly[62], Fossi Francesca[14], Francony Gilles[31], Frantzén Janek[69], Freo Ulderico[70], Frisvold Shirin[71], Furmanov Alex[72], Gagliardo Pablo[73], Galanaud Damien[25], Gao Guoyi[74], Geleijns Karin[41], Ghuysen Alexandre[75], Giraud Benoit[24], Glocker Ben[76], Gomez Pedro A.[42], Grossi Francesca[59], Gruen Russell L.[77], Gupta Deepak[78], Haagsma Juanita A.[46], Hadzic Ermin[64], Haitsma Iain[79], Hartings Jed A.[80], Helbok Raimund[21], Helseth Eirik[81], Hertle Daniel[28], Hill Sean[82], Hoedemaekers Astrid[83], Hoefer Stefan[51], Hutchinson Peter J.[1], Haberg Asta Kristine[84], Jacobs Bram[85], Janciak Ivan[86], Janssens Koen[58], Jiang Ji-yao[74], Jones Kelly[87], Kalala Jean-Pierre[88], Kamnitsas Konstantinos[76], Karan Mladen[89], Karau Jana[20], Katila Ari[69], Kaukonen Maija[90], Keeling David[52], Kerforne Thomas[24], Ketharanathan Naomi[41], Kettunen Johannes[91], Kivisaari Riku[90], Kolias Angelos G.[1], Kolumbán Bálint[92], Kompanje Erwin[93], Kondziella Daniel[67], Koskinen Lars-Owe[35], Kovács Noémi[92], Kálovits Ferenc[94], Lagares Alfonso[42], Lanyon Linda[82], Laureys Steven[95], Lauritzen Martin[67], Lecky Fiona[96], Ledig Christian[76], Lefering Rolf[97], Legrand Valerie[98], Lei Jin[61], Levi Leon[99], Lightfoot Roger[100], Lingsma Hester[46], Loeckx Dirk[101], Lozano Angels[16], Luddington Roger[17], Luijten-Arts Chantal[83], Maas Andrew I.R.[58], MacDonald Stephen[17], MacFayden Charles[65], Maegele Marc[102], Majdan Marek[33], Major Sebastian[103], Manara Alex[104], Manhes Pauline[31], Manley Geoffrey[105], Martin Didier[106], Martino Costanza[2], Maruenda Armando[16], Maréchal Hugues[55], Mastelova Dagmara[86], Mattern Julia[28], McMahon Catherine[107], Melegh Béla[108], Menon David[65], Menovsky Tomas[58], Morganti-Kossmann Cristina[109], Mulazzi Davide[38], Mutschler Manuel[102], Mühlan Holger[110], Negru Ancuta[111], Nelson David[82], Neugebauer Eddy[102], Newcombe Virginia[65], Noirhomme Quentin[95], Nyirádi József[4], Oddo Mauro[112], Oldenbeuving Annemarie[113], Oresic Matej[114], Ortolano Fabrizio[38], Palotie Aarno[91,115,116], Parizel Paul M.[117], Patruno Adriana[118], Payen

Jean-François[31], Perera Natascha[119], Perlbarg Vincent[25], Persona Paolo[120], Peul Wilco[121], Pichon Nicolas[122], Piilgaard Henning[67], Piippo Anna[90], Pili Floury Sébastien[123], Pirinen Matti[91], Ples Horia[111], Polinder Suzanne[46], Pomposo Inigo[40], Psota Marek[33], Pullens Pim[117], Puybasset Louis[124], Ragauskas Arminas[125], Raj Rahul[90], Rambadagalla Malinka[126], Rehorčíková Veronika[33], Rhodes Jonathan[127], Richardson Sylvia[128], Ripatti Samuli[91], Rocka Saulius[125], Rodier Nicolas[122], Roe Cecilie[129], Roise Olav[130], Roks Gerwin[131], Romegoux Pauline[31], Rosand Jonathan[132], Rosenfeld Jeffrey[109], Rosenlund Christina[133], Rosenthal Guy[72], Rossaint Rolf[47], Rossi Sandra[120], Rostalski Tim[110], Rueckert Daniel[76], Ruiz de Arcaute Felix[101], Rusnák Martin[86], Sacchi Marco[14], Sahakian Barbara[65], Sahuquillo Juan[134], Sakowitz Oliver[135,136], Sala Francesca[118], Sanchez-Pena Paola[25], Sanchez-Porras Renan[28,135], Sandor Janos[137], Santos Edgar[28], Sasse Nadine[51], Sasu Luminita[59], Savo Davide[118], Schipper Inger[138], Schlößer Barbara[20], Schmidt Silke[110], Schneider Annette[97], Schoechl Herbert[139], Schoonman Guus[131], Schou Rico Frederik[140], Schwendenwein Elisabeth[9], Schöll Michael[28], Sir özcan[141], Skandsen Toril[142], Smakman Lidwien[143], Smeets Dirk[101], Smielewski Peter[54], Sorinola Abayomi[144], Stamatakis Emmanuel[65], Stanworth Simon[52], Stegemann Katrin[110], Steinbüchel Nicole[145], Stevens Robert[146], Stewart William[147], Steyerberg Ewout W.[46], Stocchetti Nino[148], Sundström Nina[35], Synnot Anneliese[149,150], Szabó József[94], Söderberg Jeannette[82], Taccone Fabio Silvio[16], Tamás Viktória[144], Tanskanen Päivi[90], Tascu Alexandru[34], Taylor Mark Steven[33], Te Ao Braden[68], Tenovuo Olli[69], Teodorani Guido[151], Theadom Alice[68], Thomas Matt[104], Tibboel Dick[41], Tolias Christos[152], Tshibanda Jean-Flory Luaba[153], Tudora Cristina Maria[111], Vajkoczy Peter[154], Valeinis Egils[155], Van Hecke Wim[101], Van Praag Dominique[58], Van Roost Dirk[88], Van Vlierberghe Eline[101], Vande Vyvere Thijs[101], Vanhaudenhuyse Audrey[25,95], Vargiolu Alessia[118], Vega Emmanuel[156], Verheyden Jan[101], Vespa Paul M.[157], Vik Anne[158], Vilcinis Rimantas[159], Vizzino Giacinta[14], Vleggeert-Lankamp Carmen[143], Volovici Victor[79], Vulekovic Peter[89], Vámos Zoltán[66], Wade Derick[57], Wang Kevin K.W.[160], Wang Lei[61], Wildschut Eno[41], Williams Guy[65], Willumsen Lisette[67], Wilson Adam[5], Wilson Lindsay[161], Winkler Maren K.L.[103], Ylén Peter[162], Younsi Alexander[28], Zaaroor Menashe[99], Zhang Zhiqun[163], Zheng Zelong[28], Zumbo Fabrizio[2], de Lange Stefanie[97], de Ruiter Godard C.W.[143], den Boogert Hugo[18], van Dijck Jeroen[164], van Essen Thomas A.[121], van Heugten Caroline[57], van der Jagt Mathieu[165], van der Naalt Joukje[85]

1 Division of Neurosurgery, Department of Clinical Neurosciences, Addenbrooke's Hospital & University of Cambridge, Cambridge, UK

2 Department of Anesthesia & Intensive Care, M. Bufalini Hospital, Cesena, Italy

3 Department of Clinical Neurosciences, Addenbrooke's Hospital & University of Cambridge, Cambridge, UK

4 János Szentágothai Research Centre, University of Pécs, Pécs, Hungary

5 University of Cincinnati, Cincinnati, Ohio, United States

6 Division of Surgery and Clinical Neuroscience, Department of Physical Medicine and Rehabilitation, Oslo University Hospital and University of Oslo, Oslo, Norway

7 Department of Neurosurgery, University Hospital Northern Norway, Tromso, Norway

8 Department of Physical Medicine and Rehabilitation, University hospital Northern Norway

9 Trauma Surgery, Medical University Vienna, Vienna, Austria

10 Department of Neurosurgery, Elisabeth-Tweesteden Ziekenhuis, Tilburg, the Netherlands

11 Department of Anesthesiology & Intensive Care, University Hospital Nancy, Nancy, France

12 Riga Eastern Clinical University Hospital, Riga, Latvia

13 Raymond Poincare hospital, Assistance Publique—Hopitaux de Paris, Paris, France

14 NeuroIntensive Care, Niguarda Hospital

15 Neurointensive Care, Sheffield Teaching Hospitals NHS Foundation Trust, Sheffield, UK

16 Department Anesthesiology and Surgical-Trauma Intensive Care, Hospital Clinic Universitari de Valencia, Spain

17 Cambridge University Hospitals, Cambridge, UK

18 Department of Neurosurgery, Radboud University Medical Center

19 Department of Neurosurgery, University of Szeged, Szeged, Hungary

20 Institute for Transfusion Medicine (ITM), Witten/Herdecke University, Cologne, Germany

21 Department of Neurocritical care, Innsbruck Medical University, Innsbruck, Austria

22 Deparment of Neurosurgery & Anesthesia & intensive care medicine, Karolinska University Hospital, Stockholm, Sweden

23 NIHR Surgical Reconstruction and Microbiology Research Centre, Birmingham, UK

24 Intensive care Unit, CHU Poitiers, Poitiers, France

25 Anesthesie-Réanimation, Assistance Publique—Hopitaux de Paris, Paris, France

26 Department of Anesthesia & ICU, AOU Città della Salute e della Scienza di Torino—Orthopedic and Trauma Center, Torino, Italy

27 Department of Anesthesiology & Intensive Care, S Raffaele University Hospital, Milan, Italy

28 Department of Neurosurgery, University Hospital Heidelberg, Heidelberg, Germany

29 Department of Neurology, Odense University Hospital, Odense, denmark

30 Departments of Neuroscience and Nursing Science, Norwegian University of Science and Technology, Trondheim, Norway

31 Department of Anesthesiology & Intensive Care, University Hospital of Grenoble, Grenoble, France

32 BehaviourWorks Australia, Monash Sustainability Institute, Monash University, Victoria, Australia

33 Department of Public Health, Faculty of Health Sciences and Social Work, Trnava University, Trnava, Slovakia

34 Department of Neurosurgery, Bagdasar-Arseni Emergency Clinical Hospital, Bucharest, Romania

35 Department of Neurosurgery, Umea University Hospital, Umea, Sweden

36 Department of Neurosurgery, University of Pecs and MTA-PTE Clinical Neuroscience MR Research Group and Janos Szentagothai Research Centre, University of Pecs, Hungarian Brain Research Program, Pecs, Hungary

37 Department of Medical Psychology, Universitätsklinikum Hamburg-Eppendorf, Hamburg, Germany

38 Neuro ICU, Fondazione IRCCS Cà Granda Ospedale Maggiore Policlinico, Milan, Italy

39 Department of Epidemiology and Preventive Medicine, Monash University, Melbourne, Victoria, Australia 40 Department of Neurosurgery, Hospital of Cruces, Bilbao, Spain

41 Intensive Care and Department of Pediatric Surgery, Erasmus Medical Center, Sophia Children's Hospital, Rotterdam, The Netherlands

42 Department of Neurosurgery, Hospital Universitario 12 de Octubre, Madrid, Spain

43 Department of Neuroscience, Azienda Ospedaliera Università di Padova, Padova, Italy

44 NeuroIntensive Care, Azienda Ospedaliera San Gerardo di Monza, Monza, Italy

45 School of Medicine and Surgery, Università Milano Bicocca, Milano, Italy

46 Department of Public Health, Erasmus Medical Center-University Medical Center, Rotterdam, The Netherlands

47 Department of Anaesthesiology, University Hospital of Aachen, Aachen, Germany

48 Department of Anesthesia & Neurointensive Care, Cambridge Universiyt Hospital NHS Foundation Trust, Cambridge, UK

49 School of Public Health & PM, Monash University and The Alfred Hospital, Melbourne, Victoria, Australia

50 Radiology/MRI department, MRC Cognition and Brain Sciences Unit, Cambridge, UK

51 Institute of Medical Psycholology and Medical Sociology, Universitätsmedizin Göttingen, Göttingen, Germany

52 Oxford University Hospitals NHS Trust, Oxford, UK

53 Department of Neurosurgery, University of Pecs and MTA-PTE Clinical Neuroscience MR Research Group and Janos Szentagothai Research Centre, University of Pecs, Hungarian Brain Research Program (Grant No. KTIA 13 NAP-A-II/8), Pecs, Hungary

54 Brain Physics Lab, Division of Neurosurgery, Dept of Clinical Neurosciences, University of Cambridge, Addenbrooke's Hospital, Cambridge, UK

55 Intensive Care Unit, CHR Citadelle, Liège, Belgium

56 Intensive Care Unit, CHU, Liège, Belgium

57 Movement Science Group, Faculty of Health and Life Sciences, Oxford Brookes University, Oxford, UK

58 Department of Neurosurgery, Antwerp University Hospital and University of Antwerp, Edegem, Belgium

59 Department of Anesthesia & Intensive Care, Maggiore Della Carità Hospital, Novara, Italy

60 Department of Neurosurgery, University Hospitals Leuven, Leuven, Belgium

61 Department of Neurosurgery, Renji Hospital, Shanghai Jiaotong University School of Medicine, Shanghai, China

62 Department of Neurology, Erasmus MC, Rotterdam, the Netherlands

63 Department of Neurosurgery, Medical Faculty and clinical center University of Sarajevo, Sarajevo, Bosnia Herzegovina

64 Department of Neurosurgery, Regional Medical Center dr Safet Mujić, Mostar, Bosnia Herzegovina

65 Division of Anaesthesia, University of Cambridge, Addenbrooke's Hospital, Cambridge, UK

66 Department of Anaesthesiology and Intensive Therapy, University of Pécs, Pécs, Hungary

67 Departments of Neurology, Clinical Neurophysiology and Neuroanesthesiology, Region Hovedstaden Rigshospitalet, Copenhagen, Denmark

68 National Institute for Stroke and Applied Neurosciences, Faculty of Health and Environmental Studies, Auckland University of Technology, Auckland, New Zealand

69 Rehabilitation and Brain Trauma, Turku University Central Hospital and University of Turku, Turku, Finland

70 Department of Medicine, Azienda Ospedaliera Università di Padova, Padova, Italy

71 Department of Anesthesiology and Intensive care, University Hospital Northern Norway, Tromso, Norway

72 Department of Neurosurgery, Hadassah-hebrew University Medical center, Jerusalem, Israel

73 Fundación Instituto Valenciano de Neurorrehabilitación (FIVAN), Valencia, Spain

74 Department of Neurosurgery, Shanghai Renji hospital, Shanghai Jiaotong University/ school of medicine, Shanghai, China

75 Emergency Department, CHU, Liège, Belgium

76 Department of Computing, Imperial College London, London, UK

77 Lee Kong Chian School of Medicine, Nanyang Technological University, Singapore; and Monash University, Australia

78 Department of Neurosurgery, Neurosciences Centre & JPN Apex trauma centre, All India Institute of Medical Sciences, New Delhi-110029, India

79 Department of Neurosurgery, Erasmus MC, Rotterdam, the Netherlands

80 Department of Neurosurgery, University of Cincinnati, Cincinnati, Ohio, USA

81 Department of Neurosurgery, Oslo University Hospital, Oslo, Norway

82 Department of Physiology and Pharmacology, Section of Perioperative Medicine and Intensive Care, Karolinska Institutet, Stockholm, Sweden

83 Department of Intensive Care Medicine, Radboud University Medical Center

84 Department of Medical Imaging, St. Olavs Hospital and Department of Neuroscience, Norwegian University of Science and Technology, Trondheim, Norway

85 Department of Neurology, University Medical Center Groningen, Groningen, Netherlands

86 International Neurotrauma Research Organisation, Vienna, Austria

87 National Institute for Stroke & Applied Neurosciences of the AUT University, Auckland, New Zealand

88 Department of Neurosurgery, UZ Gent, Gent, Belgium

89 Department of Neurosurgery, Clinical centre of Vojvodina, Novi Sad, Serbia

90 Helsinki University Central Hospital

91 Institute for Molecular Medicine Finland, University of Helsinki, Helsinki, Finland

92 Hungarian Brain Research Program—Grant No. KTIA 13 NAP-A-II/8, University of Pécs, Pécs, Hungary

93 Department of Intensive Care and Department of Ethics and Philosophy of Medicine, Erasmus Medical Center, Rotterdam, The Netherlands

94 Department of Neurological & Spinal Surgery, Markusovszky University Teaching Hospital, Szombathely, Hungary

95 Cyclotron Research Center, University of Liège, Liège, Belgium

96 Emergency Medicine Research in Sheffield, Health Services Research Section, School of Health and Related Research (ScHARR), University of Sheffield, Sheffield, UK

97 Institute of Research in Operative Medicine (IFOM), Witten/Herdecke University, Cologne, Germany

98 VP Global Project Management CNS, ICON, Paris, France

99 Department of Neurosurgery, Rambam Medical Center, Haifa, Israel

100 Department of Anesthesiology & Intensive Care, University Hospitals Southhampton NHS Trust, Southhampton, UK

101 icoMetrix NV, Leuven, Belgium

102 Cologne-Merheim Medical Center (CMMC), Department of Traumatology, Orthopedic Surgery and Sportmedicine, Witten/Herdecke University, Cologne, Germany

103 Centrum für Schlaganfallforschung, Charité—Universitätsmedizin Berlin, Berlin, Germany

104 Intensive Care Unit, Southmead Hospital, Bristol, Bristol, UK

105 Department of Neurological Surgery, University of California, San Francisco, California, USA

106 Department of Neurosurgery, CHU, Liège, Belgium

107 Department of Neurosurgery, The Walton centre NHS Foundation Trust, Liverpool, UK

108 Department of Medical Genetics, University of Pécs, Pécs, Hungary

109 National Trauma Research Institute, The Alfred Hospital, Monash University, Melbourne, Victoria, Australia

110 Department Health and Prevention, University Greifswald, Greifswald, Germany

111 Department of Neurosurgery, Emergency County Hospital Timisoara, Timisoara, Romania

112 Centre Hospitalier Universitaire Vaudois

113 Department of Intensive Care, Elisabeth-Tweesteden Ziekenhuis, Tilburg, the Netherlands

114 Department of Systems Medicine, Steno Diabetes Center, Gentofte, Denmark

115 Analytic and Translational Genetics Unit, Department of Medicine; Psychiatric & Neurodevelopmental Genetics Unit, Department of Psychiatry; Department of Neurology, Massachusetts General Hospital, Boston, MA, USA

116 Program in Medical and Population Genetics; The Stanley Center for Psychiatric Research, The Broad Institute of MIT and Harvard, Cambridge, MA, USA

117 Department of Radiology, Antwerp University Hospital and University of Antwerp, Edegem, Belgium

118 NeuroIntenisve Care Unit, Department of Anesthesia & Intensive Care Azienda Ospedaliera San Gerardo di Monza, Monza, Italy

119 International Projects Management, ARTTIC, Munchen, Germany

120 Department of Anesthesia & Intensive Care, Azienda Ospedaliera Università di Padova, Padova, Italy

121 Dept. of Neurosurgery, Leiden University Medical Center, Leiden, The Netherlands and Dept. of Neurosurgery, Medical Center Haaglanden, The Hague, The Netherlands

122 Intensive Care Unit, CHU Dupuytren, Limoges, France

123 Intensive Care Unit, CHRU de Besançon, Besançon, France

124 Department of Anesthesiology and Critical Care, Pitié -Salpêtrière Teaching Hospital, Assistance Publique, Hôpitaux de Paris and University Pierre et Marie Curie, Paris, France

125 Department of Neurosurgery, Kaunas University of technology and Vilnius University, Vilnius, Lithuania

126 Rezekne Hospital, Latvia

127 Department of Anaesthesia, Critical Care & Pain Medicine NHS Lothian & University of Edinburgh, Edinburgh, UK

128 MRC Biostatistics Unit, Cambridge Institute of Public Health, Cambridge, UK

129 Department of Physical Medicine and Rehabilitation, Oslo University Hospital/University of Oslo, Oslo, Norway

130 Division of Surgery and Clinical Neuroscience, Oslo University Hospital, Oslo, Norway

131 Department of Neurology, Elisabeth-TweeSteden Ziekenhuis, Tilburg, the Netherlands

132 Broad Institute, Cambridge MA Harvard Medical School, Boston MA, Massachusetts General Hospital, Boston MA, USA

133 Department of Neurosurgery, Odense University Hospital, Odense, Denmark

134 Department of Neurosurgery, Vall d'Hebron University Hospital, Barcelona, Spain

135 Klinik für Neurochirurgie, Klinikum Ludwigsburg, Ludwigsburg, Germany

136 University Hospital Heidelberg, Heidelberg, Germany

137 Division of Biostatistics and Epidemiology, Department of Preventive Medicine, University of Debrecen, Debrecen, Hungary

138 Department of Traumasurgery, Leiden University Medical Center, Leiden, The Netherlands

139 Department of Anaesthesiology and Intensive Care, AUVA Trauma Hospital, Salzburg, Austria

140 Department of Neuroanesthesia and Neurointensive Care, Odense University Hospital, Odense, Denmark

141 Department of Emergency Care Medicine, Radboud University Medical Center

142 Department of Physical Medicine and Rehabilitation, St.Olavs Hospital and and Department of Neuroscience, Norwegian University of Science and Technology, Trondheim, Norway

143 Neurosurgical Cooperative Holland, Department of Neurosurgery, Leiden University Medical Center and Medical Center Haaglanden, Leiden and The Hague, The Netherlands

144 Department of Neurosurgery, University of Pécs, Pécs, Hungary

145 Universitätsmedizin Göttingen, Göttingen, Germany

146 Division of Neuroscience Critical Care, John Hopkins University School of Medicine, Baltimore, USA

147 Department of Neuropathology, Queen Elizabeth University Hospital and University of Glasgow, Glasgow, UK

148 Department of Pathophysiology and Transplantation, Milan University, and Neuroscience ICU, Fondazione IRCCS Cà Granda Ospedale Maggiore Policlinico, Milano, Italy

149 Australian & New Zealand Intensive Care Research Centre, Department of Epidemiology and Preventive Medicine, School of Public Health and Preventive Medicine, Monash University, Melbourne, Australia

150 Cochrane Consumers and Communication Review Group, Centre for Health Communication and Participation, School of Psychology and Public Health, La Trobe University, Melbourne, Australia

151 Department of Reahabilitation, M. Bufalini Hospital, Cesena, Italy

152 Department of Neurosurgery, Kings college London, London, UK

153 Radiology/MRI Department, CHU, Liège, Belgium

154 Neurologie, Neurochirurgie und Psychiatrie, Charité—Universitätsmedizin Berlin, Berlin, Germany

155 Pauls Stradins Clinical University Hospital, Riga, Latvia

156 Department of Anesthesiology-Intensive Care, Lille University Hospital, Lille, France

157 Director of Neurocritical Care, University of California, Los Angeles, USA

158 Department of Neurosurgery, St.Olavs Hospital and Department of Neuroscience, Norwegian University of Science and Technology, Trondheim, Norway

159 Department of Neurosurgery, Kaunas University of Health Sciences, Kaunas, Lithuania

160 Department of Psychiatry, University of Florida, Gainesville, Florida, USA

161 Division of Psychology, University of Stirling, Stirling, UK

162 VTT Technical Research Centre, Tampere, Finland

163 University of Florida, Gainesville, Florida, USA

164 Department of Neurosurgery, The HAGA Hospital, The Hague, The Netherlands

165 Department of Intensive Care, Erasmus MC, Rotterdam, the Netherlands

## Author Contributions

**Conceptualization:** Ari Ercole, David W. Nelson.

**Data curation:** Abhishek Dixit, Shubhayu Bhattacharyay.

**Formal analysis:** Ari Ercole, Abhishek Dixit, Frederick A. Zeiler.

**Funding acquisition:** Andrew I. R. Maas.

**Methodology:** Ari Ercole, Daan Nieboer, Omar Bouamra, David K. Menon, Simone A. Dijkland, Hester F. Lingsma, Lindsay Wilson, Fiona Lecky, Ewout W. Steyerberg.

**Project administration:** Shubhayu Bhattacharyay, Andrew I. R. Maas.

**Resources:** David K. Menon.

**Writing – original draft:** Ari Ercole, Shubhayu Bhattacharyay, Frederick A. Zeiler, Daan Nieboer, Omar Bouamra, Andrew I. R. Maas.

**Writing – review & editing:** Ari Ercole, David W. Nelson.

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
