## [Decision Letter · Decision Letter 0]

7 Jun 2021

Imputation strategies for missing baseline neurological assessment covariates after traumatic brain injury: a CENTER-TBI study

PONE-D-21-11931

Dear Dr. Ercole,

We’re pleased to inform you that your manuscript has been judged scientifically suitable for publication and will be formally accepted for publication once it meets all outstanding technical requirements.

Kind regards,

Corstiaan den Uil

Academic Editor

PLOS ONE

Additional Editor Comments (optional):

Reviewers' comments:

Reviewer's Responses to Questions

**Comments to the Author**

1. Is the manuscript technically sound, and do the data support the conclusions?

Reviewer #1: Yes

2. Has the statistical analysis been performed appropriately and rigorously? 

Reviewer #1: Yes

3. Have the authors made all data underlying the findings in their manuscript fully available?

Reviewer #1: Yes

4. Is the manuscript presented in an intelligible fashion and written in standard English?

Reviewer #1: Yes

5. Review Comments to the Author

Reviewer #1: Dear author,

I really liked your paper. Well structured, well written with an interesting subject.

I do think this is how scientific work should be done and this serves as an example of how to design and execute an eccelent idea.

6. PLOS authors have the option to publish the peer review history of their article (what does this mean?). If published, this will include your full peer review and any attached files.

Reviewer #1: **Yes: **Jagoš Golubović

---

## [Editor Report · Acceptance letter]

29 Jul 2021

PONE-D-21-11931 

Imputation strategies for missing baseline neurological assessment covariates after traumatic brain injury: a CENTER-TBI study 

Dear Dr. Ercole:

I'm pleased to inform you that your manuscript has been deemed suitable for publication in PLOS ONE. Congratulations! Your manuscript is now with our production department. 

Kind regards, 

on behalf of

Dr. Corstiaan den Uil 

Academic Editor

PLOS ONE